# Mechanical Properties of Flexible TPU-Based 3D Printed Lattice Structures: Role of Lattice Cut Direction and Architecture

**DOI:** 10.3390/polym13172986

**Published:** 2021-09-03

**Authors:** Victor Beloshenko, Yan Beygelzimer, Vyacheslav Chishko, Bogdan Savchenko, Nadiya Sova, Dmytro Verbylo, Andrei Voznyak, Iurii Vozniak

**Affiliations:** 1Donetsk Institute for Physics and Engineering Named after O.O. Galkin, National Academy of Sciences of Ukraine, pr. Nauki, 46, 03028 Kyiv, Ukraine; biloshenko.va@gmail.com (V.B.); yanbeygel@gmail.com (Y.B.); chishko@ukr.net (V.C.); 2Department of Applied Ecology, Technology of Polymers and Chemical Fibers, Kyiv National University of Technologies and Design, Nemirovicha Danchenko Str., 2, 03056 Kyiv, Ukraine; 1079@ukr.net (B.S.); djanc@ukr.net (N.S.); 3Institute for Problems in Materials Science I.M. Frantsevich, National Academy of Sciences of Ukraine, Krzhizhanovsky Str., 3, 03142 Kyiv, Ukraine; ver@ipms.kiev.ua; 4Department of General Technical, Disciplines and Vocational Training, Kryvyi Rih State Pedagogical University, Gagarin Av. 54, 50086 Kryvyi Rih, Ukraine; avvoznyak76@gmail.com; 5Centre of Molecular and Macromolecular Studies, Polish Academy of Sciences, Sienkiewicza Str., 112, 90363 Lodz, Poland

**Keywords:** lattice material, flexible TPU, 3D printing, internal architecture, mechanical properties

## Abstract

This study addresses the mechanical behavior of lattice materials based on flexible thermoplastic polyurethane (TPU) with honeycomb and gyroid architecture fabricated by 3D printing. Tensile, compression, and three-point bending tests were chosen as mechanical testing methods. The honeycomb architecture was found to provide higher values of rigidity (by 30%), strength (by 25%), plasticity (by 18%), and energy absorption (by 42%) of the flexible TPU lattice compared to the gyroid architecture. The strain recovery is better in the case of gyroid architecture (residual strain of 46% vs. 31%). TPUs with honeycomb architecture are characterized by anisotropy of mechanical properties in tensile and three-point bending tests. The obtained results are explained by the peculiarities of the lattice structure at meso- and macroscopic level and by the role of the pore space.

## 1. Introduction

Currently, much attention is paid to the fabrication and study of lattice (cellular) structures because of their excellent functionality due to their superior mechanical properties, large surface area, and open pores [1]. They can be used in various technological fields, including structural lightweight design [2,3], acoustic and thermal insulation [4,5,6], shock absorption [7], and as biomaterials for implants and scaffolds for tissue engineering [8,9]. Conventional technologies for the fabrication of lattice materials severely limit the possibility of forming structures with complex architecture characterized by a strictly defined shape and size of cells [10]. Many of these limitations can be overcome by using additive technologies [11,12,13].

The mechanical behavior of lattice materials, their crash resistance, and energy absorption capacity are important issues in additive manufacturing of regular cell structures. Rigid thermoplastic polymers such as ABS or PLA are mainly used to fabricate lattice polymer materials with different architectures by 3D printing [14,15,16]. The use of flexible thermoplastics for these purposes is not so widespread. This is because the 3D printing process with flexible material is more demanding and cannot be realized with most 3D printers available on the market. Besides, the determination of suitable technological parameters for flexible polymers requires additional, time-consuming optimization studies [17,18,19].

Thermoplastic polyurethane (TPU) is one of the promising polymers that can find wide practical application as a lattice material. In [20,21], it was shown that 3D printing of TPU provides a unique opportunity to fabricate customized flexible cellular structures that can be designed and optimized for specific energy absorbing applications. TPU lattice materials can be fabricated from both rigid TPU and flexible TPU and can also differ in architecture (i.e., type of cell structure). For example, for TPU honeycombs (hexagonal cells), it was shown that the energy absorption properties under compression depend on the orientation of the cells as well as the strain rate [21]. In [22], the effect of different gradient gradings produced by TPU materials on the energy absorption of honeycomb structures was demonstrated. The results showed that the energy absorption curves of these structures can be controlled by grading the density of the structures. In [23], the effects of cyclic loading and impact loading on the gradient energy absorption of honeycomb structures were investigated. The authors [17] analyzed the deformation process of regular cell structures based on the flexible TPU under quasi-static loading conditions. As shown in [24], TPU-based honeycomb structures have the potential for repeatable and high specific energy absorption, with absorption efficiency not worse than rigid polyurethane foams. Varying the cell size and wall thickness of TPU honeycombs facilitated the change in stiffness but provides only a modest opportunity to change the shape of the stress–strain curve [24].

The mechanical properties of lattice materials are also largely determined by the parameters of 3D printing (layer thickness, printing speed, raster angle, raster width, air gap etc.). The build orientation and the loading direction during the experiment also have a significant influence on the mechanical properties of additively manufactured components. For example, it was shown in [25] that TPU parts printed flat and edge oriented have better tensile strength and deformability than those printed on edge. It should be noted that the TPU samples studied did not have a lattice structure but consisted of a solid material. The authors of [20] also stated the importance and need for further research to understand the difference in elasticity and energy absorption capacity between the axial printing and transverse directions to determine whether or not 3D printing orientation has an effect on the mechanical properties of TPU-based lattice structures.

In our previous work [26], we investigated the influence of build direction and loading direction on the mechanical properties in three-point bending test of 3D-printed rigid TPU-based lattice structures. For lattice structures with square cells, it was shown that the investigated specimens are characterized by a strong anisotropy of the mechanical properties, which depends on the build direction and the loading direction. Thereby, the influence of the loading direction is significantly stronger for the samples printed vertically or at an angle of 45°, while the properties of the horizontally printed lattice structures are almost isotropic.

The main characteristic of flexible TPU is high deformability. This, combined with the right architecture of the lattice material, can lead to high efficiency in energy absorption and crashworthiness, for example, 3D-printed flexible TPU-based lattice materials also have great potential in the fabrication of lightweight, custom-shaped structures and functional parts for applications in various fields such as aerospace engineering, medical devices and sports equipment [27,28,29,30,31]. The aim of this work is to present results regarding the mechanical response of 3D printed lattice structures made of flexible TPU subjected to static (tension, three-point bending, compression) and dynamic (dynamic mechanical thermal analysis) tests. The novelty of this work lies in the complex investigation of the effects of architecture and lattice cut direction on the mechanical properties of 3D printed flexible TPU under different types of loading.

## 2. Experimental Section

### 2.1. Materials and Processing

TPU Desmopan 3690 AU (Covestro AG, Leverkusen, Germany) was used as raw material. Two types of lattice architectures were investigated: honeycomb and gyroid (Figure 1).

Models for the creation of honeycomb and gyroid architecture were created by using a program generated infill Simplify3d and Ultimaker Cura. The honeycomb structure was interpenetrating open hexagonal cells with sizes of 3 and 6 mm and a wall thickness of 0.36 mm. The corresponding gyroid structure was described by the formula:(1)sin(1.5x)·cos(1.5y) + sin(1.5y)·cos(1.5z) + sin(1.5z)·cos(1.5x)=0

They were later converted into rectangular bars by linear array duplication (Figure 1). The resulting models were saved in .stl format and sliced with the Simplify 3D package. Isometric views of individual cells of lattice structures are presented in Figure 2.

The 3D printing was done with the printer Flashforge Creator pro (Zhejiang Flashforge 3D Technology Co., Ltd., Jinhua City, China). The parameters of 3D printing were as follows: the nozzle temperature was 245 °C, the temperature of the building platform was 90 °C, the printing speed was 2000 mm/min. The filament was produced by processing of TPU in a single-screw extruder (Polymer Mash LTD, Kyiv, Ukraine; D = 25 mm, L/D = 16 and 50 rpm), followed by two-stage cooling (in water at temperatures of 60 °C and 15 °C) and drying in a heating chamber with air circulation (8 h, 50 °C). The temperatures within the zones were 190–200–210 °C. Filament production speed was 16 m/min. The diameter of the monofilament is 1.75 ± 0.05 mm.

The samples of the lattice TPU with honeycomb and gyroid architecture were printed in horizontal directions, which, as we showed in [26], provides the least anisotropy of properties. In all the cases, the structure used is self-supporting due to the small size of the open area and the sufficient bridging properties of the material and settings used for 3D printing. The thickness of the melt layers was 0.2 mm. The TPU content in the sample volume was 29.7% and 37.0% for lattice material with honeycomb and gyroid architecture, respectively. The samples were cut in two mutually perpendicular planes: perpendicular and parallel to the print direction and are designated hereinafter as Sample, type A and Sample, type B, respectively. The large side of the specimens was oriented in the x direction (Figure 1). The mechanical characteristics of materials obtained by testing samples A and B are labeled below as (z, x) and (y, x), where the first letter in parentheses indicates the normal to the sample plane, and the second indicates the direction of tension-compression.

### 2.2. Characterization

The tensile and static three-point bending tests were performed at room temperature using the universal electromechanical device UTM-100 (G.S. Pisarenko Institute for Problems of Strength of the National Academy of Sciences of Ukraine, Kyiv, Ukraine) with a speed of the crosshead movement of 10 mm/min. The distance between the outer rollers in three-point bending test was 100 mm. The samples were parallelepipeds 20 × 13 × 125 mm^3^ in size. To measure the strains, a specialized sensor for measuring large deformations (up to 500 mm) was used, the probes of which were fixed directly on the samples with the help of springs. In the experiment, the difference in the stroke of the probes was recorded. The measurement error of the sample elongation was not worse than ±0.5%. The standard deviation of the observation results was taken as a measure of measurement error. An uniaxial compression test was performed using the loading frame of an universal tensile testing machine (Instron, Model 5582, High Wycombe, UK) controlled by the Bluehill^®^ II software and a compression fixture equipped with LVDT transducer, mounted close to the specimen for precise determination of the strain. It is known that boundary effects are considered negligible when the ratio of specimen diameter to unit cell size is sufficient to represent the compressive strength of the “bulk” porous material. According to [32,33,34] this ratio should be around 3, which was chosen in the compression test. The samples were cylindrical in shape with 10 mm diameter and 15 mm thickness. The temperature of the test was 25 °C. The compression rate was 5% of the initial thickness per minute. The mean values of the mechanical characteristic were calculated from the results of testing at least 5 specimens.

The viscoelastic properties were tested on rectangular specimens, 24 mm × 10 mm × 3 mm, by dynamic mechanical thermal analysis (DMA) using the DMAQ800 (TA instrument, New Castle, DE, USA). The measurement of the tangent of the loss angle tg*δ*, the storage modulus *E’*, and the loss modulus *E”* was performed in the deformation mode of single cantilevered bending. The frequency of forced sine oscillations was 1 Hz, the temperature varied from −100 to 150 °C at the heating rate of 3 °C/min.

Differential scanning calorimetry (DSC) was performed with a device DSC Q20 (TA Instruments, New Castle, DE, USA) under heating from −60 to 140 °C at the rate of 10 °C/min. The samples with a mass of 4–5 mg were cut from lattice TPU and placed in standard aluminum pans. During the measurement the DSC cell was blown off with dry nitrogen (20 mL/min).

## 3. Results and Discussion

### 3.1. Tensile and Static Three-Point Bending Tests

Figure 3 shows the stress–strain curves of lattice flexible TPU with honeycomb and gyroid architecture during tension (loading and unloading). The average values of modulus of elasticity *E*, maximum tensile strength *σ_T_*, strain at break *ε_b_*, and residual strain *ε_r_* (after 24 h) are summarized in Table 1.

It can be seen that all stress–strain curves exhibit nonlinear behavior, regardless of the internal architecture of the sample. An anisotropy of the mechanical properties of the studied specimens is observed as a function of their orientation relative to the compression direction (Table 1). For the same orientation, the values of *E, σ_T_* and *ε_b_* are higher for lattice TPU with honeycomb architecture. In turn, the degree of anisotropy of these parameters is much lower for lattice TPU with gyroid architecture. This fact is in agreement with the results of the static three-point bending test of lattice TPU. Indeed, the modulus of elasticity for a sample with honeycomb architecture is 1.5 MPa (sample, type A) and 0.66 MPa (sample, type B), and for a sample with gyroid architecture, it is 0.72 and 0.71 MPa, respectively. It should be noted that the specimens of type A do not fail within the travel of the testing machine and show higher deformability and higher maximum tensile strength. The unloading paths show a large hysteresis loop with residual strain. After unloading, a greater strain recovery occurs with time. The residual strains measured 24 h after the tests (Table 1) also show anisotropy as a function of the lattice cut direction, indicating a more complete strain recovery for lattice TPU with a gyroid architecture.

The obtained results can be explained by the peculiarities of the multiscale structure of A- and B-type test specimens for TPUs with honeycomb and gyroid architecture. Let us consider them at the mesoscopic and macroscopic scales. The meso-scale refers to the scale where the location of each structural element is visible, which is determined by the 3D printing technology and does not depend on the architecture of the material. The macro scale refers to the scale on which the architecture of the material manifests itself.

Figure 1 shows that in the A- and B-type samples the mesostructure is different: in the first case the polymer filaments are interconnected along the width of the sample, in the second-along the thickness, which is much smaller than the width. This explains the lower strength of the B-type samples for the two studied architectures. At the macro level, the structural features of the A- and B-type samples are fundamentally different for the honeycomb and gyroid architectures. In the case of honeycomb architecture, the structures of the images of the two types are completely different, because in the XY plane this structure looks like hexagons, and in the ZX plane-layers (Figure 1). This explains the large difference in their mechanical properties (Table 1). At the same time, for a gyroid architecture at this scale, the structures of the samples of types A and B are the same, which explains the closeness of their mechanical properties. After making the change of variables in relation (1):(2)$$x^{*}=\;y,\;{\;y}^{*}=\;z,\;{\;z}^{*}=\;x\;\;$$
the following Equation (3) of the gyroid in new coordinates is obtained:(3)$$\sin\left(1.5y^{*}\right)\cdot \cos(1.5z^{*})\;+\;\sin\left(1.5z^{*}\right)\cdot \cos(1.5x^{*})\;+\;\sin\left(1.5x^{*}\right)\cdot \cos\left(1.5y^{*}\right)=0$$

A comparison of Equations (1) and (3) shows that these relations agree. It follows that the sample of the first type, cut from the gyroid in coordinates $\left(x^{*},\;{\;y}^{*},\;{\;z}^{*}\right)$, has the same architecture as the sample of the first type, cut in coordinates (x, y, z). However, the plane $\left(x^{*},\;{\;y}^{*}\right)\;$ corresponds to the plane (y, z), i.e., the sample of the first type in the new coordinates is the sample of the second type in the old coordinates. This shows that in the case of a gyroid, the samples of the first and second types have the same structure at the macroscopic level.

### 3.2. DMA and DSC Analyzes

Figure 4, as an example, shows the results obtained from DMA tests of 3D printed lattice TPU with honeycomb architecture. DMA curves for the lattice TPU with gyroid architecture are qualitatively the same. In Table 2, *E’*, *E”* are storage modulus and loss modulus at room temperature, respectively; *T_E_*, *T_tgδ_* are the temperatures of the peaks of the maxima of *E”* and tg*δ*, respectively; tg*δ* is the tangent of loss angle at room temperature; tg*δ_m_* is the maximum of tg*δ*.

The *E’(T)*, *E’’(T)* and tg*δ(T)* curves of lattice TPU show a shape typical of semicrystalline thermoplastic elastomers throughout the temperature range below the flow region [35]. The temperature-independent rubber plateau corresponds to rigid TPU segments that form stable crystals softening in a relatively narrow temperature range, which is evident from the increase in tg*δ(T)* at the end of the DMA curve. The temperature location of the maximum of the temperature dependences of *E”* and tg*δ*, which determine the glass transition of soft TPU segments, as well as the height of the maximum of tg*δ*, which characterizes the energy dissipation of a material (mechanical losses), depend weakly on the architecture and the lattice intersection direction. A similar conclusion was drawn from the DSC data (shown in the Appendix A). For all tested lattice flexible TPU samples, a *T_g_* associated with the glass transition temperatures of the soft TPU segments was observed at the temperatures around −22 °C. At the same time, the storage modulus and the loss modulus are higher in the case of lattice TPU with honeycomb architecture in the same lattice cut direction.

In general, the dynamic mechanical analysis data (Table 2) are in good agreement with the results of the tensile and static three-point bending tests, which show similar regularities. The comparison of viscoelastic properties of honeycomb architecture lattice rigid TPU [26] and lattice flexible TPU showed that a decrease in matrix rigidity naturally leads to a significant (up to 10-fold) decrease in *E’* and *E’’* values.

### 3.3. Compression Test

Theoretically, anisotropic structures are not conducive to energy absorption because under uncertain loading conditions, the energy absorption in the weak direction of the structure will be extremely low. Figure 5 shows the stress-strain curves of lattice TPU specimens of different architectures under compression. In both cases, three typical stages of compression of such materials are observed: the linear-elastic stage, the plastic plateau stage, and the densification stage, when all cells collapse, and the material deforms like a continuous medium. It is important to note that the studied samples were not destroyed-they were completely compressed, and over time they almost completely restore their shape (shown in the Appendix A). The modulus of elasticity, plateau stress *σ_L,_* and densification strain *ε_D_* of the lattice TPU structures are given in Table 3.

In contrast to the above results of the tensile and three-point bending tests, the compression test shows an almost complete absence of anisotropy of the mechanical properties (modulus of elasticity) for both the lattice TPU with a honeycomb and gyroid architecture (Table 3). This behavior of the materials under compression, in contrast to bending and tension, can be explained by the fact that in this case the properties are largely determined by a reduction in the volume of the pore space. In terms of its influence on the properties of the materials, this factor outweighs the role of the architecture, which explains the closeness of the mechanical properties of the type A and B specimens.

Regardless of the lattice section direction, the TPU lattice with a honeycomb architecture is characterized by a higher strain hardening rate in both small and large deformations (Figure 5), higher values of modulus of elasticity, plateau stress (Table 3) and energy absorption (Figure 6) compared to the TPU lattice with a gyroid architecture. With the increase of compressive strain, several plateaus appeared corresponding to a gradual layer-by-layer collapse (Figure 5, inset). The value of onset densification strain *ε_D_* of the lattice structures, where the compaction region begins (which is the practical limit for energy absorption applications), is slightly lower for samples with a honeycomb architecture than for samples with a gyroid architecture (Table 3).

The energy absorption (*EA*) capacity of lattice materials can be calculated from the area under the stress–strain curve:(4)$$EA=\int_{0}^{\varepsilon_{D}}\sigma \left(\varepsilon \right)d\varepsilon$$
where *σ* is stress and *ε* is strain. The specific energy absorption (*SEA*) is the energy absorption per unit mass which is used to evaluate the energy-absorbing efficiency. The *SEA* is defined as:(5)$$SEA=\;\frac{EA}{m}$$
where *m* is the mass of the lattice material, *EA*-is the absorbed deformation energy per unit volume.

By integrating *σ(ε)* at different strains, the energy absorption of the lattice structure under different strains can be obtained, as shown in Figure 6a. It can be seen that at the initial stage, the energy absorption capacity of the studied samples is practically the same. However, when the strain exceeds 0.3, the energy absorption of the lattice TPU with honeycomb architecture is higher than that of the gyroid TPU. The differences in energy absorption between the two lattice cut directions correlate with those mentioned earlier and amount to about 25% for both lattice TPU with honeycomb and gyroid architectures. The data from SEA show that the energy absorption performance is significantly higher for lattice TPU with honeycomb architecture, regardless of the lattice cut direction (Figure 6b).

## 4. Conclusions

The presented study considers the possibility of 3D printing flexible TPU lattice materials with two types of lattice architectures: honeycombs and gyroids. The effects of lattice cut direction and architecture on the mechanical response under various types of loading are systematically investigated. The main findings from this study are as follows.

It is shown that compared to the gyroid architecture, the honeycomb architecture determines the TPU lattice with 30%, 25% and 18% higher values of *E*, *σ_T_*, *ε_b_*, respectively, measured in both tension and compression.

The presence of the anisotropy of the mechanical properties of the TPU lattice in the case of tension and three-point bending and its absence during compression are noted. The effect of the lattice cut direction is much stronger for samples with honeycomb architecture, while the properties of gyroid-based lattice structures are almost isotropic. The characteristics of the mechanical behavior of the studied TPU lattices are associated with differences in their structure at the meso and macro levels, as well as with the leading role of the pore space in compression tests.

The energy absorption capacity of a lattice structures depends on its architecture and lattice cut direction. The energy absorption performance (by 42%) is significantly higher for lattice TPU with honeycomb architecture, regardless of the lattice cut direction.

Flexible TPU-based 3d printed lattice materials have broad prospects for use in many areas of everyday life, including in medicine, furniture, automotive, civil engineering, etc. Their undoubted advantage is lightness, high mechanical, energy absorption and elastic recovery properties. The results from this study provide experimental data for analyzing and optimizing products made of 3D printed flexible TPU.

## Figures and Tables

**Figure 1 polymers-13-02986-f001:**
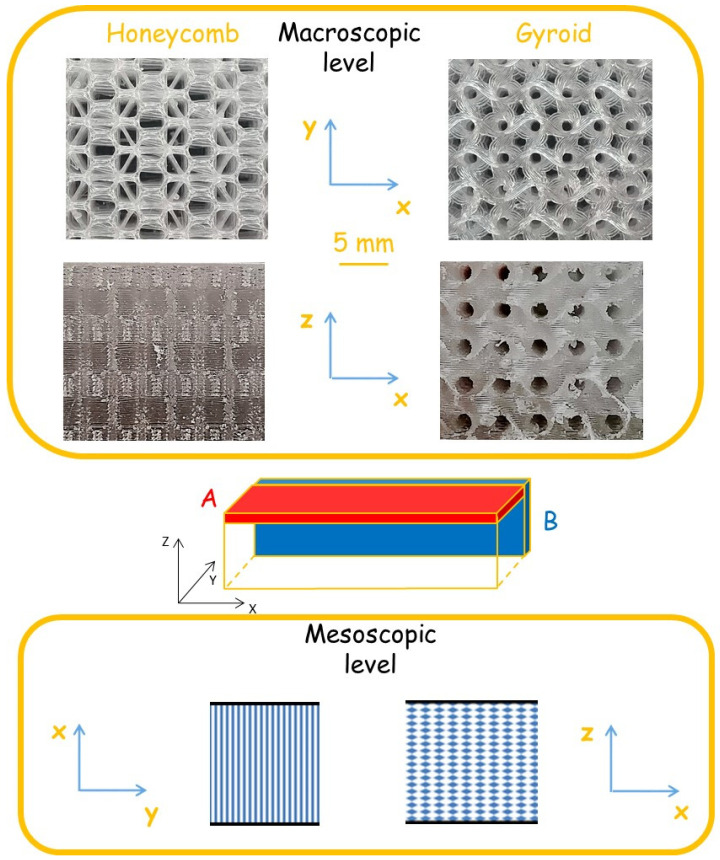
The peculiarities of the multiscale structure of flexible TPU lattice materials with honeycomb and gyroid architecture as well as the arrangement of A- and B-type test specimens.

**Figure 2 polymers-13-02986-f002:**
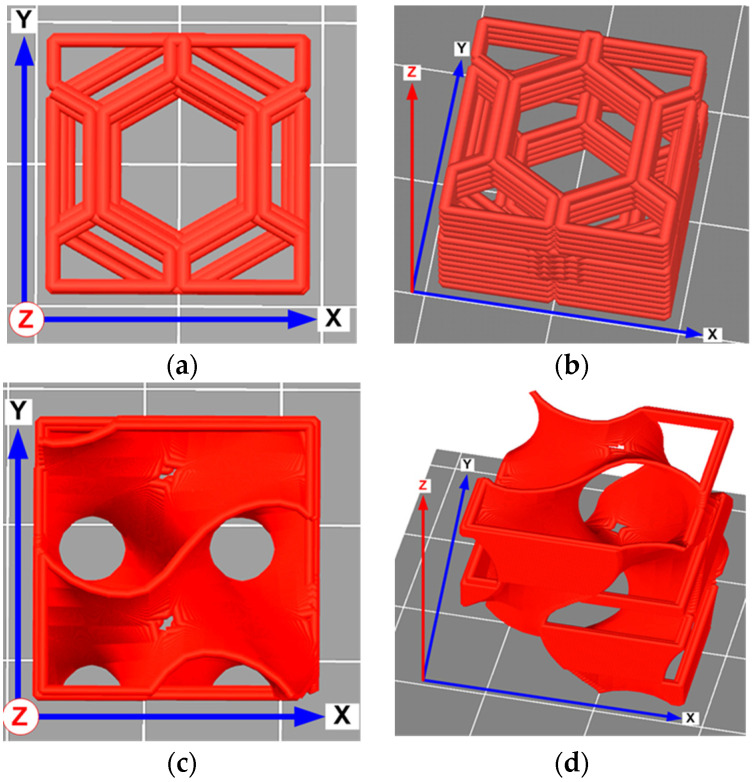
Isometric views of individual cells of honeycomb (**a**,**b**) and gyroid (**c**,**d**) architectures.

**Figure 3 polymers-13-02986-f003:**
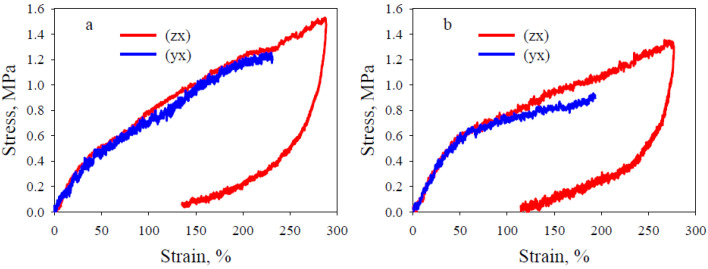
Representative stress–strain curves of lattice flexible TPU under tension. (**a**) honeycomb, (**b**) gyroid architecture.

**Figure 4 polymers-13-02986-f004:**
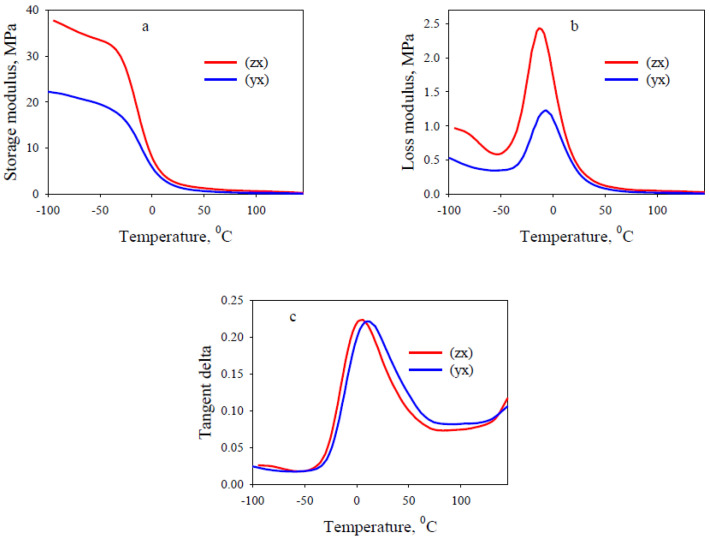
DMA curves for lattice flexible TPU with honeycomb architecture. (**a**)–*E’(T)*, (**b**)–*E’’(T)*, (**c**)–tg*δ(T)* dependencies.

**Figure 5 polymers-13-02986-f005:**
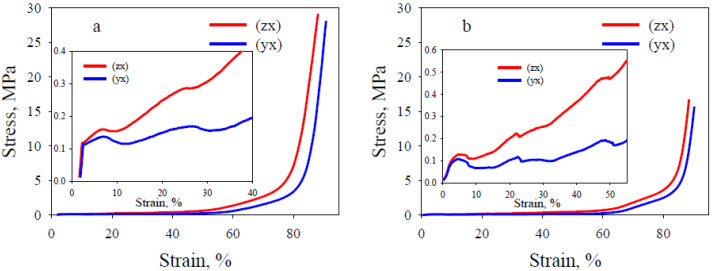
Representative stress–strain curves of lattice flexible TPU under compression. (**a**)–honeycomb, (**b**)–gyroid architecture. The inset shows the enlarged initial part of the same curves.

**Figure 6 polymers-13-02986-f006:**
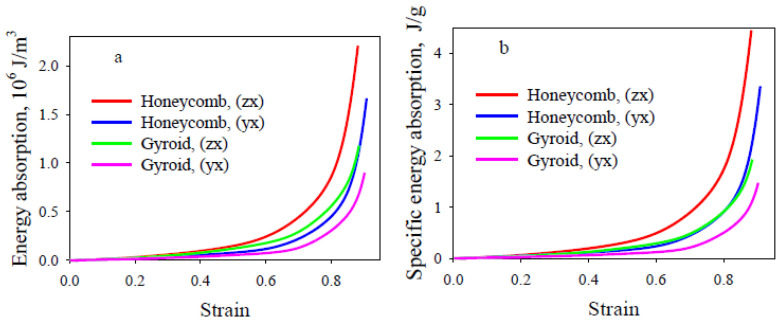
Energy absorption (**a**) and specific energy absorption (**b**) of the lattice flexible TPU structures.

**Table 1 polymers-13-02986-t001:** Mechanical properties of lattice flexible TPU. Tensile test.

Internal Architecture	Sample Orientation	*E*, MPa	*σ_T_*, MPa	*ε_b_*, %	*ε_r_*, %
Honeycomb	(zx)	2.1 ± 0.1	Not break	>280	46 ± 2
	(yx)	1.6 ± 0.1	1.26 ± 0.05	230 ± 5	21 ± 2
Gyroid	(zx)	1.5 ± 0.1	Not break	>280	31 ± 2
	(yx)	1.4 ± 0.1	0.94 ± 0.05	190 ± 6	12 ± 1

**Table 2 polymers-13-02986-t002:** Viscoelastic properties of lattice flexible TPU.

Internal Architecture	Sample Orientation	*E’*, MPa	*E’’*, MPa	*E’’m*, MPa	*T_E_*, °C	*T_tgδ_*, °C	tg*δ*	tg*δ_m_*
Honeycomb	(zx)	2.5	0.40	2.6	−13	7	0.18	0.23
	(yx)	1.7	0.35	1.4	−11	6	0.19	0.23
Gyroid	(zx)	0.7	0.16	1.1	−12	6	0.2	0.27
	(yx)	0.8	0.15	0.85	−11	5	0.2	0.25

**Table 3 polymers-13-02986-t003:** Mechanical properties of lattice flexible TPU. Compression test.

Internal Architecture	Sample Orientation	*E*, MPa	*σ_L_*, MPa	*ε_y_*, %	*ε_D_*, %
Honeycomb	(zx)	10.8 ± 0.3	4.5 ± 0.2	1.6 ± 0.1	79 ± 3
	(yx)	10.7 ± 0.2	4.3 ± 0.1	1.4 ± 0.1	84 ± 4
Gyroid	(zx)	7.5 ± 0.3	4.1 ± 0.1	1.3 ± 0.1	83 ± 3
	(yx)	7.3 ± 0.3	4.0 ± 0.1	1.1 ± 0.2	86 ± 3

## Data Availability

All the data will be available to the readers.

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
