# Peer review of "Mechanical Properties of Flexible TPU-Based 3D Printed Lattice Structures: Role of Lattice Cut Direction and Architecture"

_polymers, 2021, doi:10.3390/polym13172986_

Round 1

Reviewer 1 Report

Dear Authors,

Remarks:

  1. Point 2.1 „Two types of lattice architectures were investigated: honeycomb and gyroid.” – I propose to add the reference to Figure 1.
  2. equations 1 and 3. Commas should be changed to dots.
  3. point 2.2. “The measurement of the tangent of the loss angle tgδ, the” and later in the text (section 3.2) – tg should not be italic
  4. point 2.2. “from -100 to 150°C” – please check and verify the temperature range
  5. Table 1. Description of the units should be corrected (MPa)
  6. Text/Table 1, 2, 3/ Figure 2/etc. Anisotropic materials have different modulus of elasticity. In the paper there is only one designation of this material parameter (E), however, two samples with different orientations are tested. This should be corrected and more information about directions of the measured/calculated variables/parameters should be provided (i.e. by subscripts in the particular designations).
  7. Equation 2. Such designation generally means derivative of a function. I propose to use different letters instead “ ’ ”.
  8. Figure 3. There is no description of the figures.
  9. Page 8 “which is evident from the increase in tan δ(T) at the end of the DMA curve” – please verify (tan).
  10. The conclusions should be more precise

Kind Regards,

Author Response

Rewiever 1

Dear Authors,

Remarks:

Point 2.1 „Two types of lattice architectures were investigated: honeycomb and gyroid.” – I propose to add the reference to Figure 1.

We have added the reference to Figure 1 in this sentence.

equations 1 and 3. Commas should be changed to dots.

Commas have been changed to dots in the equations 1 and 3

point 2.2. “The measurement of the tangent of the loss angle tgδ, the” and later in the text (section 3.2) – tg should not be italic

We have removed italics in tgd

point 2.2. “from -100 to 150°C” – please check and verify the temperature range

This temperature range is correct

Table 1. Description of the units should be corrected (MPa)

Description of the units has been corrected

Text/Table 1, 2, 3/ Figure 2/etc. Anisotropic materials have different modulus of elasticity. In the paper there is only one designation of this material parameter (E), however, two samples with different orientations are tested. This should be corrected and more information about directions of the measured/calculated variables/parameters should be provided (i.e. by subscripts in the particular designations).

We have added the required information to section 2.1

The large side of the specimens was oriented in the x direction (Fig. 1). The mechanical characteristics of materials obtained by testing samples A and B are labeled below as (z, x) and (y, x), where the first letter in parentheses indicates the normal to the sample plane, and the second indicates the direction of tension-compression. For example: E (z, x), E (y, x) are the elastic moduli of samples A and B, respectively.

Equation 2. Such designation generally means derivative of a function. I propose to use different letters instead “ ’ ”.

We have used an asterisk instead of a dash

Figure 3. There is no description of the figures.

There is a description of Figure 3 in the article: for example:..The E'(T), E''(T) and tgδ(T) curves of lattice TPU show a shape typical of semicrystalline thermoplastic elastomers throughout the temperature range below the flow region [25]. ..The temperature location of the maximum of the temperature dependences of Е" and tgδ, which determine the glass transition of soft TPU segments, as well as the height of the maximum of tgδ, which characterizes the energy dissipation of a material (mechanical losses), depend weakly on the architecture and the lattice intersection direction. ....At the same time, the storage modulus and the loss modulus are higher in the case of lattice TPU with honeycomb architecture in the same lattice cut direction.

Page 8 “which is evident from the increase in tan δ(T) at the end of the DMA curve” – please verify (tan).

The increase in tan δ (T) at the end of the DMA curve is indeed observed experimentally and is associated with softening of stable TPU crystals.

The conclusions should be more precise

We have added specifics in the conclusions

The presented study considers the possibility of 3D printing flexible TPU lattice materials with two types of lattice architectures: honeycombs and gyroids. It is shown that compared to the gyroid architecture, the honeycomb architecture determines the TPU lattice with 30%, 55%, 25% and 18% higher values of Е, σy, σT, εb, respectively, measured in both tension and compression. Higher EA and SEA values by 42% and 43%, respectively, are also observed in the case of honeycomb architecture. The presence of the anisotropy of the mechanical properties of the TPU lattice in the case of tension and three-point bending and its absence during compression are noted. The TPU lattice with honeycomb architecture has more pronounced anisotropy. The characteristics of the mechanical behavior of the studied TPU lattices are associated with differences in their structure at the meso and macro levels, as well as with the leading role of the pore space in compression tests.

Rewiever 2

The paper deals with quasi-static testing and characterization of TPU material 3D printed lattices with different unit-cell geometry. Production material characterization is provided additionally to the properties of the investigated lattices. I recommend the paper for publication after a minor revision after the following comments are addressed by the authors:

- There is a typo in the title (a dash between the last two words). The paper definitely needs to be revised by a native English speaker to correct grammar issues.

The paper has been checked with the InTExt program and the grammar issues have been revised.

- The paper needs to be better graphically documented. Event though there are photographs(?) of the investigated lattices provided in Fig. 1, I miss the images of mechanical testing setups with emplaced specimens and particularly the images of the whole samples used in the tests, preferably including the visualization from the parametric modeler used for their design. This should also include a visualization of individual unit-cells.

A description of mechanical test setups has been added to section 2.2. Isometric views of individual cells of lattice structures are presented in Fig.2. in the revised manuscript

The tensile and static three-point bending tests were performed at room temperature using the universal electromechanical device UТМ-100 with a speed of the crosshead movement of 10 mm/min. The distance between the outer rollers in three-point bending test was 100 mm. To measure the strains, a specialized sensor for measuring large deformations (up to 500 mm) was used, the probes of which were fixed directly on the samples with the help of springs. In the experiment, the difference in the stroke of the probes was recorded. The measurement error of the sample elongation was not worse than ± 0.5%. The standard deviation of the observation results was taken as a measure of measurement error. Uniaxial compression test was performed using the loading frame of an universal tensile testing machine (Instron, Model 5582) controlled by the Bluehill® II software and a compression fixture equipped with LVDT transducer, mounted close to the specimen for precise determination of the strain. It is known that boundary effects are considered negligible when the ratio of specimen diameter to unit cell size is sufficient to represent the compressive strength of the “bulk” porous material. According to [23-25] this ratio should be around 3, which was chosen in the compression test. The temperature of the test was 25 °C. The compression rate was 5% of the initial thickness per minute. The mean values of the mechanical characteristic were calculated from the results of testing at least 5 specimens.

- The mathematical formulae seem to be formatted as an ordinary text - proper mathematical typesetting style should be used.

The mathematical formulae have been reformatted

- On P. 3, you claim that the printing speed during build process of the samples was 2000 mm/s, which I find as an unusually high value, while the datasheet of the printing devices shows 10 - 100 mm/s. Is this an error, or the actual device was tuned to different parameters?

It was a typo. In the revised version of the paper, the true value has bene given

- Provide more detailed caption of Fig. 1. Include scale bar in its upper part. What is the purpose of the 3D coordinate system triads in its lower part?

More detailed caption of Fig. 1. as well as a scale bar have been provided. 3D coordinates are replaced with 2D coordinates.

Figure 1. The peculiarities of the multiscale structure of flexible TPU lattice materials with honeycomb and gyroid architecture as well as the arrangement of A- and B-type test specimens.

- I miss the information regarding dimensions of the samples and dimensions of the unit-cells.

Dimensions of the samples and dimensions of the unit-cells are given in section 2.1.

- P. 6, Tab. 1:

- There are some cyrillic characters.

Description of the units has been corrected

- I miss information on how the mechanical test have been performed and how the strains have been assessed. This has serious impact on reliability of results and this point has to be addressed.

The missing information on the description of mechanical test methods has been added.

2.2. Characterization

The tensile and static three-point bending tests were performed at room temperature using the universal electromechanical device UТМ-100 with a speed of the crosshead movement of 10 mm/min. The distance between the outer rollers in three-point bending test was 100 mm. To measure the strains, a specialized sensor for measuring large deformations (up to 500 mm) was used, the probes of which were fixed directly on the samples with the help of springs. In the experiment, the difference in the stroke of the probes was recorded. The measurement error of the sample elongation was not worse than ± 0.5%. The standard deviation of the observation results was taken as a measure of measurement error. Uniaxial compression test was performed using the loading frame of an universal tensile testing machine (Instron, Model 5582) controlled by the Bluehill® II software and a compression fixture equipped with LVDT transducer, mounted close to the specimen for precise determination of the strain. It is known that boundary effects are considered negligible when the ratio of specimen diameter to unit cell size is sufficient to represent the compressive strength of the “bulk” porous material. According to [23-25] this ratio should be around 3, which was chosen in the compression test. The temperature of the test was 25 °C. The compression rate was 5% of the initial thickness per minute. The mean values of the mechanical characteristic were calculated from the results of testing at least 5 specimens.

    - By comparing the data provided in the Tab. 1 and the graphs in Fig. 2, I do not find the mutual relationship between the values of individual mechanical characteristics and the representative stress-strain curves.

    We removed the true values, replacing them with engineering ones to avoid confusion.

- P. 10, Tab. 3:

- The comment regarding the details on mechanical tests and strain evaluation applies here, too, together with the comparison of data in Tab. 3 with representative graphs in Fig. 4 and their relationship.

Comments have been added to the description of mechanical test methods. See edited text for section 2.2

    - In both tables, you present elastic characteristics derived from the loading curve as I understand the manuscript. How did you deal with boundary effects present in such porous constructs and how did you performed the correction on stiffness and errors introduced by the loading device itself?

Since the sensor for measuring large deformations had the counterweights of the probes, and the friction in the bearing blocks was negligible, then the measurement error of the specimen elongation was no worse than ± 0.5%.The standard deviation of the observation results was taken as a measure of measurement error.

It is known that boundary effects are considered negligible when the ratio of specimen diameter to unit cell size is sufficient to represent the compressive strength of the “bulk” porous material. According to [23-25] this ratio should be around 3, which was chosen in the compression test.

- The discussion section is completely missing.

We decided not to separate the discussion section, but to leave the previous version of the presentation of the results and discussion, adding some new data.

- You provided repeated comments on recovery of the samples after loading. Provide more details including graphs of these processes.

We have added the corresponding results to the Supplementary Materials section, Fig. 2.

Rewiever 3

There are several recommendations to the present paper:

polymer MDPI journal template should be used

Polymer MDPI journal template has been used

Reviewer 2 Report

The paper deals with quasi-static testing and characterization of TPU material 3D printed lattices with different unit-cell geometry. Production material characterization is provided additionally to the properties of the investigated lattices. I recommend the paper for publication after a minor revision after the following comments are addressed by the authors:

- There is a typo in the title (a dash between the last two words). The paper definitely needs to be revised by a native English speaker to correct grammar issues.

- The paper needs to be better graphically documented. Event though there are photographs(?) of the investigated lattices provided in Fig. 1, I miss the images of mechanical testing setups with emplaced specimens and particularly the images of the whole samples used in the tests, preferably including the visualization from the parametric modeler used for their design. This should also include a visualization of individual unit-cells.

- The mathematical formulae seem to be formatted as an ordinary text - proper mathematical typesetting style should be used.

- On P. 3, you claim that the printing speed during build process of the samples was 2000 mm/s, which I find as an unusually high value, while the datasheet of the printing devices shows 10 - 100 mm/s. Is this an error, or the actual device was tuned to different parameters?

- Provide more detailed caption of Fig. 1. Include scale bar in its upper part. What is the purpose of the 3D coordinate system triads in its lower part?

- I miss the information regarding dimensions of the samples and dimensions of the unit-cells.

- P. 6, Tab. 1:
    - There are some cyrillic characters.
    - I miss information on how the mechanical test have been performed and how the strains have been assessed. This has serious impact on reliability of results and this point has to be addressed.
    - By comparing the data provided in the Tab. 1 and the graphs in Fig. 2, I do not find the mutual relationship between the values of individual mechanical characteristics and the representative stress-strain curves.

- P. 10, Tab. 3:
    - The comment regarding the details on mechanical tests and strain evaluation applies here, too, together with the comparison of data in Tab. 3 with representative graphs in Fig. 4 and their relationship.
    - In both tables, you present elastic characteristics derived from the loading curve as I understand the manuscript. How did you deal with boundary effects present in such porous constructs and how did you performed the correction on stiffness and errors introduced by the loading device itself?

- The discussion section is completely missing.

- You provided repeated comments on recovery of the samples after loading. Provide more details including graphs of these processes.

Author Response

Rewiever 2

The paper deals with quasi-static testing and characterization of TPU material 3D printed lattices with different unit-cell geometry. Production material characterization is provided additionally to the properties of the investigated lattices. I recommend the paper for publication after a minor revision after the following comments are addressed by the authors:

- There is a typo in the title (a dash between the last two words). The paper definitely needs to be revised by a native English speaker to correct grammar issues.

The paper has been checked with the InTExt program and the grammar issues have been revised.

- The paper needs to be better graphically documented. Event though there are photographs(?) of the investigated lattices provided in Fig. 1, I miss the images of mechanical testing setups with emplaced specimens and particularly the images of the whole samples used in the tests, preferably including the visualization from the parametric modeler used for their design. This should also include a visualization of individual unit-cells.

A description of mechanical test setups has been added to section 2.2. Isometric views of individual cells of lattice structures are presented in Fig.2. in the revised manuscript

The tensile and static three-point bending tests were performed at room temperature using the universal electromechanical device UТМ-100 with a speed of the crosshead movement of 10 mm/min. The distance between the outer rollers in three-point bending test was 100 mm. To measure the strains, a specialized sensor for measuring large deformations (up to 500 mm) was used, the probes of which were fixed directly on the samples with the help of springs. In the experiment, the difference in the stroke of the probes was recorded. The measurement error of the sample elongation was not worse than ± 0.5%. The standard deviation of the observation results was taken as a measure of measurement error. Uniaxial compression test was performed using the loading frame of an universal tensile testing machine (Instron, Model 5582) controlled by the Bluehill® II software and a compression fixture equipped with LVDT transducer, mounted close to the specimen for precise determination of the strain. It is known that boundary effects are considered negligible when the ratio of specimen diameter to unit cell size is sufficient to represent the compressive strength of the “bulk” porous material. According to [23-25] this ratio should be around 3, which was chosen in the compression test. The temperature of the test was 25 °C. The compression rate was 5% of the initial thickness per minute. The mean values of the mechanical characteristic were calculated from the results of testing at least 5 specimens.

- The mathematical formulae seem to be formatted as an ordinary text - proper mathematical typesetting style should be used.

The mathematical formulae have been reformatted

- On P. 3, you claim that the printing speed during build process of the samples was 2000 mm/s, which I find as an unusually high value, while the datasheet of the printing devices shows 10 - 100 mm/s. Is this an error, or the actual device was tuned to different parameters?

It was a typo. In the revised version of the paper, the true value has bene given

- Provide more detailed caption of Fig. 1. Include scale bar in its upper part. What is the purpose of the 3D coordinate system triads in its lower part?

More detailed caption of Fig. 1. as well as a scale bar have been provided. 3D coordinates are replaced with 2D coordinates.

Figure 1. The peculiarities of the multiscale structure of flexible TPU lattice materials with honeycomb and gyroid architecture as well as the arrangement of A- and B-type test specimens.

- I miss the information regarding dimensions of the samples and dimensions of the unit-cells.

Dimensions of the samples and dimensions of the unit-cells are given in section 2.1.

- P. 6, Tab. 1:

- There are some cyrillic characters.

Description of the units has been corrected

- I miss information on how the mechanical test have been performed and how the strains have been assessed. This has serious impact on reliability of results and this point has to be addressed.

The missing information on the description of mechanical test methods has been added.

2.2. Characterization

The tensile and static three-point bending tests were performed at room temperature using the universal electromechanical device UТМ-100 with a speed of the crosshead movement of 10 mm/min. The distance between the outer rollers in three-point bending test was 100 mm. To measure the strains, a specialized sensor for measuring large deformations (up to 500 mm) was used, the probes of which were fixed directly on the samples with the help of springs. In the experiment, the difference in the stroke of the probes was recorded. The measurement error of the sample elongation was not worse than ± 0.5%. The standard deviation of the observation results was taken as a measure of measurement error. Uniaxial compression test was performed using the loading frame of an universal tensile testing machine (Instron, Model 5582) controlled by the Bluehill® II software and a compression fixture equipped with LVDT transducer, mounted close to the specimen for precise determination of the strain. It is known that boundary effects are considered negligible when the ratio of specimen diameter to unit cell size is sufficient to represent the compressive strength of the “bulk” porous material. According to [23-25] this ratio should be around 3, which was chosen in the compression test. The temperature of the test was 25 °C. The compression rate was 5% of the initial thickness per minute. The mean values of the mechanical characteristic were calculated from the results of testing at least 5 specimens.

    - By comparing the data provided in the Tab. 1 and the graphs in Fig. 2, I do not find the mutual relationship between the values of individual mechanical characteristics and the representative stress-strain curves.

    We removed the true values, replacing them with engineering ones to avoid confusion.

- P. 10, Tab. 3:

- The comment regarding the details on mechanical tests and strain evaluation applies here, too, together with the comparison of data in Tab. 3 with representative graphs in Fig. 4 and their relationship.

Comments have been added to the description of mechanical test methods. See edited text for section 2.2

    - In both tables, you present elastic characteristics derived from the loading curve as I understand the manuscript. How did you deal with boundary effects present in such porous constructs and how did you performed the correction on stiffness and errors introduced by the loading device itself?

Since the sensor for measuring large deformations had the counterweights of the probes, and the friction in the bearing blocks was negligible, then the measurement error of the specimen elongation was no worse than ± 0.5%.The standard deviation of the observation results was taken as a measure of measurement error.

It is known that boundary effects are considered negligible when the ratio of specimen diameter to unit cell size is sufficient to represent the compressive strength of the “bulk” porous material. According to [23-25] this ratio should be around 3, which was chosen in the compression test.

- The discussion section is completely missing.

We decided not to separate the discussion section, but to leave the previous version of the presentation of the results and discussion, adding some new data.

- You provided repeated comments on recovery of the samples after loading. Provide more details including graphs of these processes.

We have added the corresponding results to the Supplementary Materials section, Fig. 2.

Reviewer 3 Report

There are several recommendations to the present paper:

  • polymer MDPI journal template should be used

Author Response

There are several recommendations to the present paper:

polymer MDPI journal template should be used

Polymer MDPI journal template has been used

Round 2

Reviewer 1 Report

Dear Authors,

Remarks:

  1. Figure 3 – missing line in legend (type B).
  2. Tables 1-3, Figures 3-5 – there is still no information about directions of the stresses and material properties with respect to the local coordinate system defined in Figure 1.
  3. Figure 2 – missing coordinate systems.
  4. Different formatting of the figures is used. It should be corrected.
  5. Figure 4. - there is no description of the figures a-b-c. I propose to add symbols for particular variables in descriptions of the axes.
  6. All figures- please correct decimal separators.
  7. Missing equation numbering in section 3.3

Kind Regards,

Author Response

Remarks:

    Figure 3 – missing line in legend (type B).

We have added the missing line.

Tables 1-3, Figures 3-5 – there is still no information about directions of the stresses and material properties with respect to the local coordinate system defined in Figure 1.

We have corrected the position of the axes of the coordinate system in Fig. 1, and they now correctly show the sample orientation for testing. In addition, in Tables 1-3 and Figures 3-5, we have changed the designation "type of samples" with a designation for its orientation (z, x) and (y, x), including the normal to the plane of the cut (first letter) and the direction of the load (second letter).

    Figure 2 – missing coordinate systems.

We have added the missing coordinate systems.

    Different formatting of the figures is used. It should be corrected.

The drawings were all made using the SigmaPlot program, the size of the symbols and the line thickness are the same.

    Figure 4. - there is no description of the figures a-b-c. I propose to add symbols for particular variables in descriptions of the axes.

We have added the missing description.

 Figure 4. DMA curves for lattice flexible TPU with honeycomb architecture. (a) – E'(T), (b) – E''(T), (c) – tgδ(T) dependencies.

    All figures- please correct decimal separators.

Unfortunately, we do not know how the decimal separator can be changed in the Sigmaplot program. We will turn to Polymers technical support for help.

    Missing equation numbering in section 3.3

We have added the numbering in section 3.3

Reviewer 3 Report

The present paper deals with the investigation of the mechanical properties of the 3D printed TPU samples.

Comments:

1) Abstract should present the results of the investigations and must be specific avoiding general sentences.

2) up to date literature about additive manufacturing and optimization of the mechanical properties should be referenced. For example, in polymers mdpi: https://doi.org/10.3390/polym12123009; many others also exist. 

3) Drawing speed of filament is missing.

4) filament diameter is missing.

5) "mesoscopic and macroscopic scale" therm should be referenced in the literature. If it is the original author's terminology - more explanation should be provided. What is the scale -m, cm, mm, microns, nm ?

6) shape of DMA samples is missing, shape of samples for compression tests are also misisng

7) Could you discuss why Tg of the TPU polymer is different from the printed shape? Tg is an intrinsic chemical/physical characteristic of the material.

8) lines:

  1. a Tg associated with the glass transition temperatures of the soft TPU seg-

  2. ments was observed at the temperatures around -22°Ð¡

DSC and DMA showed that Tg is much higher. It is not clear.

9) Discussion of DSC measyrements is confusing. Calorimetric properties are intrinsic properties of the materials, they are shape independent. Are ther any differences in printing conditions of Gyroid and Honeicomb structures? 

10) How yield point σy calculated?

11) EA and SEA are interconnected parameters. They behave similar, just different absolute values. Only one should be shown in Figure 6. 

12) Application of the present research result should be mentioned in the conclusions.

Author Response

Comments:

  • Abstract should present the results of the investigations and must be specific avoiding general sentences.

We have added some specific data to the abstract

This study addresses the mechanical behavior of lattice materials based on flexible TPU with honeycomb and gyroid architecture fabricated by 3D printing. Tensile, compression and three-point bending tests were chosen as mechanical testing methods. The honeycomb architecture was found to provide higher values of rigidity (by 30%), strength (by 25%), plasticity (by 18%) and energy absorption (by 42%) of the flexible TPU lattice compared to the gyroid architecture. The strain recovery is better in the case of gyroid architecture (residual strain of 46% vs 31%). TPUs with honeycomb architecture are characterized by anisotropy of mechanical properties in tensile and three-point bending tests. The obtained results are explained by the peculiarities of the lattice structure at meso- and macroscopic level and by the role of the pore space.

  • up to date literature about additive manufacturing and optimization of the mechanical properties should be referenced. For example, in polymers mdpi: https://doi.org/10.3390/polym12123009; many others also exist.

We have updated the literature on additive manufacturing and optimization of the mechanical properties

3) Drawing speed of filament is missing.

We have added this information to the text. Filament production speed was 16 m/min.

4) filament diameter is missing.

We have added this information to the text. The diameter of the monofilament is 1.75 ± 0.05 mm.

5) "mesoscopic and macroscopic scale" therm should be referenced in the literature. If it is the original author's terminology - more explanation should be provided. What is the scale -m, cm, mm, microns, nm ?

These are standard terms. When applied to architectural materials, they are used, for example, in the article [Yuri Estrin, Yan Beygelzimer, Roman Kulagin, Peter Gumbsch, Peter Fratzl, Yuntian Zhu & Horst Hahn (2021) Architecturing materials at mesoscale: some current trends, Materials Research Letters, 9 : 10, 399-421, DOI: 10.1080 / 21663831.2021.1961908]

6) shape of DMA samples is missing, shape of samples for compression tests are also missing

We have added this information to the text.

DMTA measurements were carried out on rectangular specimens, 24 mm × 10 mm × 3 mm using a DMTA DMA Q 800 TA Instruments (USA) apparatus at the frequency of 1 Hz and at the heating rate of 2  C min 1.

Uniaxial compression tests were done by using specimens of 10 mm diameter and 15 mm high.

7) Could you discuss why Tg of the TPU polymer is different from the printed shape? Tg is an intrinsic chemical/physical characteristic of the material.

In the manuscript, line 232-233, we just point out that the glass transition temperature is the same for the two investigated types of lattice (“... depend weakly on the architecture and the lattice intersection direction.”). The observed difference in 1 0C is within the experimental error.

8) lines:

a Tg associated with the glass transition temperatures of the soft TPU segments was observed at the temperatures around -22°Ð¡ DSC and DMA showed that Tg is much higher. It is not clear.

The difference in the absolute values of the glass transition temperature determined by these two methods is indeed high. This could be attributed to both the fact that a porous, non-solid sample was used in the DMA experiments (see, for example, K. A. Ross , O. H. Campanella & M. R. Okos (2002) The effect of porosity on glass transition measurement, International Journal of Food roperties, 5:3, 611-628, DOI: 10.1081/JFP-120015496) and fuzziness of the Tg transition on the corresponding DSC curves.

9) Discussion of DSC measyrements is confusing. Calorimetric properties are intrinsic properties of the materials, they are shape independent. Are ther any differences in printing conditions of Gyroid and Honeicomb structures?

When forming different architecture of lattice samples, flow as well as vitrification conditions (for example, associated with a difference in the rate of heat removal) can differ and thus affect possible orientation/relaxation processes of macromolecules. In this case, the independence of the glass transition temperature from the architecture of the lattice materials indicates that the flow and vitrification conditions were either similar or their difference was insignificant.

10) How yield point σy calculated?

Yield Strength – The yield strength of the plastic is the where the material begins to deform in a plastic fashion. Prior to the yield strength, the material will act elastically meaning that if the strain were halted at any point in the elastic portion, the material would return to its original length. Once the yield strength of the plastic is attained, the material will not return to its original length and will yield.

Since the studied materials were restored elastically after the removal of the load until the moment of densification, we removed the data on the yield strength.

11) EA and SEA are interconnected parameters. They behave similar, just different absolute values. Only one should be shown in Figure 6.

We can move Fig. 6b to the supplementary materials section. however if the reviewer does not mind, for the convenience of the readers, we would like to leave it in the main body of the article.

12) Application of the present research result should be mentioned in the conclusions.

We have added this information.

The presented study considers the possibility of 3D printing flexible TPU lattice materials with two types of lattice architectures: honeycombs and gyroids. It is shown that compared to the gyroid architecture, the honeycomb architecture determines the TPU lattice with 30%, 25% and 18% higher values of Е, σT, εb, respectively, measured in both tension and compression. Higher EA and SEA values by 42% and 43%, respectively, are also observed in the case of honeycomb architecture. The presence of the anisotropy of the mechanical properties of the TPU lattice in the case of tension and three-point bending and its absence during compression are noted. The TPU lattice with honeycomb architecture has more pronounced anisotropy. The characteristics of the mechanical behavior of the studied TPU lattices are associated with differences in their structure at the meso and macro levels, as well as with the leading role of the pore space in compression tests. Flexible TPU-based 3d printed lattice materials have broad prospects for use in many areas of everyday life, including in medicine, furniture, automotive, civil engineering, etc. Their undoubted advantage is lightness, high mechanical, energy absorption and elastic recovery properties.